# Skin Cancer Detection and Classification Using Neural Network Algorithms: A Systematic Review

**DOI:** 10.3390/diagnostics14040454

**Published:** 2024-02-19

**Authors:** Pamela Hermosilla, Ricardo Soto, Emanuel Vega, Cristian Suazo, Jefté Ponce

**Affiliations:** Escuela de Ingeniería Informática, Pontificia Universidad Católica de Valparaíso, Avenida Brasil 2241, Valparaíso 2362807, Chileemanuel.vega@pucv.cl (E.V.); cristian.suazo.j@mail.pucv.cl (C.S.); jefte.ponce.h@mail.pucv.cl (J.P.)

**Keywords:** machine learning (ML), deep learning (DL), convolutional neural networks (CNNs), skin cancer, melanoma, cancer datasets

## Abstract

In recent years, there has been growing interest in the use of computer-assisted technology for early detection of skin cancer through the analysis of dermatoscopic images. However, the accuracy illustrated behind the state-of-the-art approaches depends on several factors, such as the quality of the images and the interpretation of the results by medical experts. This systematic review aims to critically assess the efficacy and challenges of this research field in order to explain the usability and limitations and highlight potential future lines of work for the scientific and clinical community. In this study, the analysis was carried out over 45 contemporary studies extracted from databases such as Web of Science and Scopus. Several computer vision techniques related to image and video processing for early skin cancer diagnosis were identified. In this context, the focus behind the process included the algorithms employed, result accuracy, and validation metrics. Thus, the results yielded significant advancements in cancer detection using deep learning and machine learning algorithms. Lastly, this review establishes a foundation for future research, highlighting potential contributions and opportunities to improve the effectiveness of skin cancer detection through machine learning.

## 1. Introduction

In the last decade, the detection and classification of skin diseases, with a particular focus on oncology, has been a trending topic within the machine learning field. In this context, among the main advancements, a relevant amount of research has been conducted on skin cancer diagnosis by the employment of deep learning techniques. The objectives and designs behind the state-of-the-art proposed approaches aimed from simple image comparison [1,2,3] to the adoption of advanced optimization methods, such as Harris’s hawk optimization [4]. Similarly, quantitative imaging biomarkers have been examined in the context of metastatic melanoma and immunological treatments [5]. However, to date, several efforts need to be carried out to prevent this highly mortal disease, as skin cancer is considered the most prevalent and dangerous type in medical oncology. It can result from previous exposure to radiation therapy or carcinogenic substances.

In order to fully understand the several scopes that define this pathology and the respective research evolution, different points need to be addressed and illustrated. Firstly, a deeper insight into the skin cancer detection process requires outlining the fundamental steps and crucial aspects involved. In this context, detection begins with a meticulous visual analysis of the skin conducted by dermatologists, which is complemented by medical evaluation of one’s family history, along with the yielded exam results, supported by techniques such as dermatoscopy. Among the most employed initial diagnostic methods is the ABCDE criteria [3,6,7], which are described in Table 1. Nevertheless, while this simple, noninvasive, and expertise-based observational approach often leads to an accurate diagnosis of between 60 and 90% of malignant tumors [3], there are scenarios where the certainty of cancer’s presence can only be ascertained through a biopsy.

Thus, it is pertinent to note that, despite the guidelines provided by the employment of the ABCDE method, the task of visual skin inspection can be inherently complex. This complexity stems from the difficulty in identifying certain melanoma characteristics. This challenge is highlighted in Figure 1, which illustrates a randomized selection of images from the ISIC 2017 dataset (International Skin Imaging Collaboration), encompassing both melanoma (“mel”) cases and other non-melanoma skin lesions (“nomel”). The resemblance between some of these images highlights the inherent difficulties in accurately distinguishing between malignant and benign skin lesions.

Secondly, given the aforementioned situation, hybrid designs become relevant. These designs are characterized by the integration and interaction of multiple disciplines, with the aim of achieving improvements and positive outcomes [8]. Therefore, it is crucial to highlight the key aspects of artificial intelligence (AI), as shown in Figure 2, which include well-known attributes and various fields that have demonstrated promising results in identifying distinctive features of malignancy in skin lesions. Hybrid-based architectures have been increasingly recognized in research, notably in [4], where a dual deep learning algorithm merged image and audio analysis through CNNs. One CNN evaluates raw images for malignancy [9], while another processes audio from image sonification, applying this dual approach to various image types.

Thirdly, it is relevant to understand the evolution behind image processing, which includes how the given algorithms and models have adequate capabilities to tackle both modern and well-known issues. For instance, “Dermo-DOCTOR” [10] addresses class homogeneity and heterogeneity, while modular camera systems automate lesion detection and tracking [11]. Also, noninvasive methods offer biopsy alternatives for assessing cellular atypia in keratinocyte cancers [12]. Innovations include CNNs combined with the Water strider algorithm for melanoma [13] and the Inception network’s use for binary classification, which was improved by transfer learning and RMSProp optimization [14]. Additionally, a U-Net based CNN and one-class SVM tackled speckle noise and subjective OCT image interpretation [15].

In general, the articles reviewed not only highlighted the evolution of image processing and machine learning techniques but also emphasized improvements in disease detection, diagnosis, and classification. AI has made significant contributions to breast cancer detection and monitoring, where techniques like logistic regression, decision trees, and CNNs have achieved up to 99% accuracy, showcasing AI’s potential in early detection and treatment, especially in regions like Asia [16]. On the other hand, they illustrated extended applications, such as Alzheimer’s and retinal disease diagnosis through CNNs, improving medical image analysis and patient outcomes.

Regarding systematic reviews in the area of skin cancer, relevant contributions were found between 2019 and 2023, covering between 19 and 100 articles [17,18,19]. In this scenario, different aims were achieved, such as works presenting a full disease-focused introduction for new research focused on the diagnosis process and generalization of advanced methodologies for detecting multiple types of cancer. Complementing the information provided, the objective of this study is focused on the review and analysis of skin cancer detection algorithms, seeking to identify and evaluate emerging trends, challenges, and opportunities in this field of research. The selection of scientific articles considered a broad range of methodologies and practical applications, enabling a comprehensive understanding of the evolution and current state of skin cancer detection technologies. Moreover, special attention was paid to evaluating the effectiveness, accuracy, and feasibility of different algorithms in various clinical and environmental contexts. This study also includes a critical analysis of the current limitations in skin cancer detection research, proposing areas for future research and development. The importance of interdisciplinarity is emphasized, involving perspectives from dermatology, oncology, informatics, and artificial intelligence to effectively address the challenges of skin cancer detection.

The purpose of this systematic review is to explore and provide answers to the research questions (RQ) that directed this study, which are defined as follows:RQ 1: What are the types of algorithms applied to detect skin cancer?RQ 2: What types of optimizers are used to improve the accuracy of the results?RQ 3: What are the datasets used in skin cancer detection studies?RQ 4: What metrics are used to validate the effectiveness of these algorithms in skin cancer detection?

Subsequently, this study includes a comparative analysis of different algorithms, evaluating not only their individual performance but also their effectiveness in comparison to each other. This will allow identification of the specific strengths and weaknesses of each method and their suitability for various clinical situations. Furthermore, the integration of these algorithms into real healthcare settings will be considered, examining challenges such as the interpretation of results by healthcare professionals and the integration of these tools into existing clinical workflows. On the other hand, this study will also explore how advancements in related technologies, such as artificial intelligence and deep learning, are influencing the development of new skin cancer detection algorithms, opening up possibilities for more accurate and efficient methods. Special attention will be paid to the evolution of diagnostic accuracy over time and how these technological advancements can contribute to improving the early detection and treatment of skin cancer. Finally, this study will seek to provide a guide for future research in the field, highlighting areas that still need development and opportunities for innovation. This will include recommendations for addressing current limitations and suggestions for future lines of research that could lead to significant improvements in the detection and treatment of skin cancer.

## 2. Materials and Methods

### 2.1. Methodology

The research methodology established for this review seeks to address the core questions that guide this study, which are framed within the investigation and response to the previously stated research questions (RQs). To effectively tackle these questions, we identified two stages. In the first stage, we focused on defining the key concepts that would aid in selecting the set of articles to undergo review. In the second stage, the research team analyzed the most significant aspects related to machine learning algorithms in skin cancer detection. In Figure 3, a representation of the proposed research methodology is presented, where the stages with their respective tools, activities, and outcomes can be identified. Through examination of the reviewed articles, relevant aspects were established concerning the result accuracy, utilized optimizers, validation metrics, and datasets employed in the studies. This allowed us to establish categories for comparing their scope and contributions while interpreting the similarities and differences among them.

### 2.2. Search Strategy

To address the research questions mentioned, this study was organized by first describing the methodology with which this review was conducted, which included everything from the collection and selection of articles based on the PRISMA 2020 reference (Preferred Reporting Items for Systematic Reviews and Meta-Analyses) [20]. This guide lays out a series of recommendations for presenting systematic reviews and meta-analyses in the scientific literature, providing a set of essential elements that should be included in the preparation and presentation of systematic reviews and meta-analyses.

For the selection of scientific articles, the WoS and Scopus databases were used, conducting various queries to search for articles that reviewed the relevant concepts, named key search terms (KSTs) for this study that contributed to answering the research questions presented, and that provided important background information related to skin cancer detection. In Figure 4 search queries (SQs) were defined to guide searches in the mentioned databases.

After defining the queries, initial searches were carried out in the WoS and Scopus databases. These searches yielded a significant number of articles, surpassing the anticipated analytical capacity of this study. Confronted with this information, strict selection criteria were implemented based on thematic relevance, the quality of the publications, and their publication dates. The process involved refining the information to generate a manageable dataset. The criteria used to determine which papers were included or excluded in this review are presented in Table 2.

Considering the initially defined queries and the previously specified eligibility criteria, a summary of the results obtained from the searches conducted in WoS and Scopus is provided below. Table 3 presents a summary of the number of articles found before and after applying inclusion and exclusion criteria.

Afterward, Mendeley Reference Manager was used to eliminate duplicates from each consulted database. Subsequently, an additional refinement was carried out by selecting articles from the Q1 and Q2 quartiles. This refers to quartiles that categorize journals based on their impact factor. The impact factor is a measure that reflects the average number of citations to recent articles published in a particular journal. Journals in quartiles Q1 and Q2 have the highest impact factors and are generally well-regarded and influential within their fields. The details of the results after applying these refinements are shown in Table 4.

Finally, a Python algorithm was deployed to merge the articles into a unified list, which was then sorted by the number of citations. The description of this is as follows: 17 works corresponded to Scopus articles, and 28 were from WoS. Also, it should be noted that the final search was performed between the second and third weeks of April, and thus the research was biased toward the articles found up to that moment. To synthesize the search process we carried out, a scheme in Figure 5 outlines the search and paper selection process, following the PRISMA 2020 framework for this systematic review.

### 2.3. Analysis Categories

To conduct the review of the selected articles, it was crucial to define the analysis categories that would be used in the context of this research. These categories and subcategories would be developed based on the key questions formulated in the study. The primary purpose of these categories was to guide and structure the analysis process, focusing it on the search for specific answers. This in turn would enable a more efficient organization of the gathered information and streamline the extraction of relevant conclusions. Figure 6 provides an outline that aligns the research questions of this study with their corresponding analysis categories.

In order to conduct this systematic review, we identified four essential categories of analysis in the previous framework. These categories were the types of algorithms employed for classification, model optimizers used within the training process, skin datasets available for research, and evaluation metrics used to quantify the performance of algorithms or models. In summary, these four categories covered the most relevant aspects for conducting analysis related to algorithms in the context of skin-related issues. Each of these categories played a key role in understanding and evaluating the algorithms used in this context. In Table 5, each category and subcategory are identified, defined, and described.

## 3. Results

In this section, we illustrate the analysis process, synthesis, and points of view regarding the state of the art reviewed. Firstly, a generalized perspective is taken in order to achieve an overview of the demographic and temporal coverage of the issues. Subsequently, the main findings identified from the analysis of the previously mentioned categories are pointed out, also establishing cross-cutting aspects in the reviewed articles in terms of the subcategories identified. Finally, a summary of this section is provided, highlighting the most relevant findings within this research.

### 3.1. Main Features of the Selected Articles

In Figure 7, a comprehensive geographic mapping is shown to emphasize the efforts and contributions made in the literature regarding skin cancer detection. The analysis revealed that the United States, India, Germany, Saudi Arabia, and the United Kingdom are the leading countries in this field. Various factors, such as health policies, skin cancer incidence rates, and investment in research infrastructure were taken into account during the selection process for the fully updated state of the art review. In total, 45 articles were selected.

The United States’ leadership with nine articles showcases the country’s advanced medical research capabilities, particularly in the development of diagnostic technologies such as dermatoscopy, imaging, and machine learning algorithms for early detection. India’s substantial representation with eight articles may indicate a strategic emphasis on technology-driven research for skin cancer detection, which is crucial in a country where access to dermatologists may be limited. Germany, Saudi Arabia, and the United Kingdom, each with seven articles, indicated their significant investments in medical technology and research. These contributions likely reflect advanced research on imaging techniques, molecular diagnostics, and the integration of artificial intelligence in clinical practice for earlier and more accurate skin cancer detection. Furthermore, the text aims to highlight the latest efforts in skin cancer detection.

Furthermore, to highlight the latest efforts in skin cancer detection, in Figure 8, we illustrate the amount of research proposed per year, which can be interpreted as this topic being a trend and a hot subject within the research community.

The increase in skin cancer detection research in 2019 highlights a global trend toward adopting artificial intelligence and machine learning, noted for enhancing diagnostic accuracy. This surge reflects the growing awareness of skin cancer as a major health issue and the potential of technology in early detection. The decline in 2021 may be attributed to the redirection of research resources due to the COVID-19 pandemic’s impact on funding and priorities, or it could represent the cyclical nature of research dynamics. The slight recovery in 2022 indicates a possible resumption of activities or the advent of new detection technologies. The lower publication count in 2023 might not fully represent ongoing research, possibly due to the year’s incompleteness or publication delays. This trend underscores the necessity of continuous research investment and highlights how global events can influence research productivity. It stresses the importance of resilience in the research community to persistently address long-term health challenges like skin cancer.

### 3.2. Findings per Analysis Category

In this section, we present a deep analysis organized by the topics illustrated in Table 5 based on the most relevant findings achieved within the literature review. In this regard, Table 6 provides a summary which identifies several reported algorithms, contributions, and output metrics.

#### 3.2.1. Types of Algorithms

In this section, we present a comprehensive overview based on the algorithms implemented and reported, which has been designed by means of CNNs, SVMs, and other hybrid methods. In this context, we can observe a clear predominance of employment of CNN algorithms, which have contributed novel approaches, such as a three-step cascade design for automatic skin lesion detection [34], fully convolutional neural network (FCN) architectures [60], a new epidermis segmentation technique [66], data augmentation plus the use of a “k-fold” cross-validation technique [71], and the aggregation of new dense pooling layers for lesion region segmentation in skin images [72]. On the other hand, interesting results have been reported by hybrid approaches, such as the extraction of deep features from pretrained networks [57] and the combination of results from each step of feature extraction [69].

This contextualization emphasizes the diversity, richness, and profitability behind the state-of-the-art approaches proposed, demonstrating the adaptability and versatility necessary to address specific challenges associated with melanoma detection. Thus, through this variety of methodology, we aim to ensure comprehensive coverage that allows for a thorough and comparative evaluation of the effectiveness of different approaches in the field of skin cancer detection.

#### 3.2.2. Model Optimizers

Regarding the model optimizers employed, this work addresses the challenge of local minima stagnation in model training, emphasizing advancements in optimization strategies. Stochastic gradient descent (SGD) with restarts has been highlighted as a solution for improving skin cancer detection accuracy [36]. The adaptability and effectiveness of Adam and SGD optimizers in various CNN architectures were demonstrated in [40,43,53], with RMSProp discussed as an alternative in [36,65,72]. The versatility of the Adam optimizer has been showcased across different neural network architectures, including ResNet-152 [55], and its utility in enhancing training efficiency through dynamic learning rate adjustments was noted in [59]. Comparative analyses, such as in [62], extend the optimizer discussion by comparing Adadelta and Adam, while the authors of [65] explored the impact of learning rate schedulers and dropout percentages. The preference for Adam due to its simplicity and effectiveness was emphasized in [71], and its strategic role in addressing with an activation function as a RelU was illustrated in [58].

#### 3.2.3. Skin Datasets

The body of research covered in the literature illustrates significant reliance over diverse datasets, particularly emphasizing the importance of specialized collections for advancing the fields of dermatology and medical imaging. Key among these are the HAM10000, ISIC, and PH2 datasets, which are extensively used for training and validating AI-driven diagnostic models [30,31,33,50,54]. The links for accessing the mentioned online datasets are provided in Table 7.

Firstly, the HAM10000 dataset concerns a large collection of dermatoscopic images that has been employed by numerous studies [46,48]. This dataset provides a diverse range of skin lesion images, making it an invaluable resource for developing algorithms capable of identifying a wide array of skin conditions. Its comprehensive nature allows for the creation of robust models that are well versed in recognizing various dermatological issues [55,62]. Secondly, the International Skin Imaging Collaboration (ISIC) archive, which concerns another pivotal resource that is frequently cited within the literature [43,47], offers an extensive repository of high-quality dermoscopic images, focusing on melanoma and other cutaneous skin conditions. The depth and breadth of the ISIC archive make it an ideal dataset for training deep learning models, particularly in the realm of melanoma diagnosis [54,59]. Thirdly, the PH2 dataset has been utilized in skin lesion analysis and has provided valuable data for assessing various types of skin lesions [69,74]. This dataset improves the diversity and depth of data available for research, contributing significantly to advancements in the field.

Regarding the science community’s appreciation of the usage of datasets, the HAM10000, ISIC, and PH2 datasets are not only valuable for their size and quality but also for their role in fostering open and collaborative research. Their availability to the research community enables a broad spectrum of studies, from fundamental image classification to complex pattern recognition tasks. This openness is crucial in facilitating advancements in medical imaging and in the application of AI in dermatology. In addition to prominent datasets such as HAM10000, the ISIC, and PH2, the research field has greatly benefited from a variety of other specialized data collections. In this regard, datasets like MED-NODE, used for developing automated diagnostic algorithms [36], and DermIS-DermQuest, providing a wealth of clinical images for dermatological analysis [41,45], have been integral to advancements in this area. Dermofit, another valuable resource offering high-quality images for lesion segmentation, further contributes to the depth of available research materials [40,53]. In the same context, the ALL-IDB2 dataset needs to be considered, mostly because it has been a key asset in skin cancer research studies [53], and Xiangya-Derm, with its extensive collection of clinical skin disease images [70] which have significantly contributed with automated diagnostic techniques. Similarly, MoleMap’s unique focus on teledermatology imaging and the comprehensive dermatoscopic images from the HAM dataset have also played crucial roles in recent research endeavors [55]. Thus, these datasets, with their unique features and specialized content, have enabled more detailed exploration and understanding of various skin conditions, improving the accuracy and efficacy of AI-based diagnostic models and paving the way for future research in dermatology and medical imaging.

#### 3.2.4. Evaluation Metrics

Evaluation metrics in skin cancer research have played a crucial role in revealing fundamental aspects. The balance between recall (sensitivity) and specificity is emphasized, underscoring the need for accurate detection of positive cases and the correct identification of negatives. The comparison between the effectiveness of CNNs and dermatologists’ expertise, as reported in [30], highlighted the importance of collaboration between artificial intelligence and human expertise. Additionally, the use of standard metrics in studies like [32] is presented as a common practice to provide a clear assessment of the model’s accuracy.

The following equations illustrate the standard metrics and measures used in the calculation of the model evaluation metrics:(1)$$\mathrm{Accuracy}=\frac{TP+TN}{\mathrm{Total}\;\mathrm{Samples}}$$
(2)$$\mathrm{Precision}=\frac{TP}{TP+FP}$$
(3)$$\mathrm{Specificity}=\frac{TN}{TN+FP}$$
(4)$$\mathrm{Recall}=\frac{TP}{TP+FN}$$
(5)$$AUC=\int_{0}^{1}\mathrm{ROC}\;\mathrm{Curve}\left(t\right)\;dt$$
where TP represents the true positives and illustrates the correctly predicted instances of the positive class. FP corresponds to the false positives, which denotes the incorrectly predicted instances of the positive class when they belong to the negative class. TN corresponds to the true negatives, where the model correctly predicted instances of the negative class. Lastly, FN represents the false negatives (FNs), which are incorrectly predicted instances of the negative class when they actually belong to the positive class.

In the literature, the evaluation process carried out over a model has been illustrated as an arduous and high complexity task. For instance, the accuracy measurement indicated by classical metrics such as TPs was presented as a fundamental approach to assessing the efficiency of deep learning models [32]. Also, the need to balance the recall (sensitivity) and specificity, as addressed in [31], was crucial to minimizing FPs or FNs in accurate skin cancer diagnosis, highlighting the relevance of achieving balance within this process. In the same context, other studies, such as [36,37], incorporated metrics like the receiver operating characteristic (ROC) curve and the quality of image reconstructions, respectively, to evaluate the model discrimination and visual representation fidelity, carrying out a diagnosis. The involvement of pathologists in [38] emphasizes the need to evaluate models in real clinical environments, highlighting accuracy and reliability.

### 3.3. Summary

Exploring the skin cancer detection field, which includes algorithms, strategies, and hybrid methods, has revealed a rich tapestry, with the employment of CNNs taking a clear lead spot within the literature. Notable mentions include the innovative three-step cascade design and architectures like the FCN, showcasing adaptability for effective melanoma identification. Also, hybrid approaches such as deep feature extraction and combining results from multiple steps have demonstrated promise for robust skin cancer detection models. These findings highlight opportunities for future studies to unravel the synergies between ensemble methods and specific CNN architectures.

By tackling the intricacies of model optimization, this review sheds light on the strategic employment of SGD with restarts in order to navigate the local minima during training. The ubiquitous Adam optimizer, favored for its simplicity and efficiency, takes the spotlight. However, the exploration of alternatives like RMSProp opens avenues for future investigations into their applicability. Optimizer selection’s role in overcoming challenges tied to activation functions suggests a fertile ground for research on optimization techniques to enhance CNN model efficiency for skin cancer detection.

Regarding the revision of the dataset, the reliance on HAM10000, ISIC, and PH2 underscores their pivotal role in advancing dermatology. Future studies could benefit from delving into specialized datasets like MED-NODE, DermIS, DermQuest, ALL-IDB2, Xiangya-Derm, and MoleMap. The collaborative and open nature of these datasets emphasizes their significance in research, urging continual diversification for comprehensive exploration. Future research might explore how dataset characteristics influence model performance and generalizability.

In the realm of evaluation metrics, this review underscores the delicate balance between recall and specificity in skin cancer research. Standard metrics like true positives (TPs), false positives (FPs), true negatives (TNs), and false negatives (FNs), alongside the AUC, accuracy, precision, specificity, and recall, play crucial roles. Future studies could refine and standardize these metrics, considering the intricate nature of skin cancer diagnosis. The collaboration between artificial intelligence and human expertise, as indicated by the inclusion of pathologists in the evaluation process, calls for future research to explore such partnerships in real clinical settings. The incorporation of classical metrics, ROC curves, and novel metrics sets the stage for nuanced evaluation approaches in future studies on skin cancer detection.

## 4. Discussion

Considering the findings of this review in relation to the analyzed studies, it is possible to point out that there are several machine learning approaches, especially those implemented with CNNs, which have been employed independently and in a hybrid manner with other techniques. These approaches have led to significant and noteworthy results in terms of model evaluation metrics for skin cancer detection. These studies provide a high level of precision, which improves the potential applications as an important complement to the work carried out by experts in the field.

To comprehensively address the final scope of this research, we refer to the questions that guided this study. These questions provide a clear framework for the aspects to be addressed and the objectives to be achieved. By answering these questions, we aim to gain a deeper and more complete understanding of the subject matter, which will enable us to recognize the implications of the mentioned findings and propose directions for future work in the field. Subsequently, a detailed account of the implications of the findings identified in each of the questions guiding this research is provided.

What are the types of algorithms applied to detect skin cancer?According to the reviewed studies, the most commonly used algorithms in the field of this research are convolutional neural networks (CNNs). In general terms, CNNs are characterized by having a series of convolutional layers, responsible for extracting the primary features by combining various kernels to generate feature maps. Additionally, pooling layers gradually reduce the image size, refining the precision of distinctive features that will be used to train the model. These layers are applied sequentially, starting from the original input image in the first layer of the network.The architecture of a CNN can vary in terms of the number of convolutional or pooling layers, and it can also incorporate fully connected (FC) layers. FC layers process prior features for classification, initiating a classification phase that may be repeated in subsequent layers, continuing the classification process until data are prepared for the final output and classification [75,76]. These layers can be complemented with activation layers to introduce nonlinearity. Additionally, techniques such as “dropout” are employed to prevent overfitting by randomly deactivating certain neurons during training. “Early stopping” enhances performance by identifying the model’s equilibrium point and halting training when it no longer learns, optimizing the use of computational resources. In summary, these techniques complement the architecture, improving classification and mitigating overfitting issues.It was also evident that the application of CNNs in conjunction with pretrained models for skin cancer detection, especially melanoma, has achieved improved accuracy results. Some of these predefined algorithms reviewed in the studies include ResNeXt, SeResNeXt, DenseNet, Xception, AlexNet, ResNet, SVMs, and random forests [31,33,43,45,60,74]. Furthermore, in the examined literature, it is pertinent to note a scarce amount of detailed documentation information concerning the software libraries utilized for algorithm implementation, a detail that warrants further investigation for future research endeavors. However, within the scope of the reviewed studies, TensorFlow and PyTorch were identified as the predominant libraries for the development and training of neural networks. The following graph illustrates the presence of the mentioned algorithms in the articles that are part of this research, which have been considered part of the proposed experimentation or a general reference in the presented conceptual framework [65,77,78,79].Figure 9 illustrates the distribution of various machine learning algorithms used instead of CNNs. The majority of applications were focused on advanced algorithms such as ResNet and SVMs, which together represented more than half of the total. Algorithms like AlexNet and DenseNet also had a significant share. Conversely, methods such as ResNeXt, SeResNeXt, and random forests were applied to a lesser extent, indicating a preference for more established or possibly more effective models in the set of applications under consideration.On the other hand, the idea of approaches based on the interaction between pretrained models with CNNs have been relevant within the literature. In Figure 10, we illustrate a comparison between purely pretrained models (orange) and hybrid pretrained models (blue). In this regard, hybrid designs led by algorithms such as ResNet have been recurrently complemented by CNNs, mostly due to the good results and affinity when working together.At the same time, the use of hybrid algorithms led to the concepts of transfer learning and fine-tuning, which are interesting techniques to explore in these areas since the main idea is to consider the use of a pretrained model and make fine adjustments for a specific task.The former refers to a technique where a model developed for one task is reused as the starting point for a model in a second task. Often, the initial layers of the pretrained network are frozen. This means that the weights of these layers are not updated during further training. The frozen layers act as generic feature extractors.The latter is an additional step in transfer learning. After initializing a model with weights from a pretrained model, training continues on the new dataset, finely adjusting the weights of some or all layers and allowing the model to more specifically adapt to the characteristics of the new dataset, which can result in better performance for a particular situation [1,8,76,80].CNNs applied in conjunction with optimization algorithms constitute a powerful approach in the field of deep learning, particularly in computer vision tasks such as image recognition, classification, and segmentation.Optimization algorithms are also essential for efficiently training a CNN. During training, the goal is to minimize a loss function, which measures how far the model’s predictions are from the actual outcomes. In this sense, optimization seeks to adjust the weights of the neural network in an attempt to reduce the loss function. Among the most used algorithms are the following:Stochastic gradient descent (SGD): This is a classic method that performs updates after viewing each training sample or a small batch of samples, making it more robust against local minima [43].Momentum: This improves SGD by taking into account the gradient from the previous update to smooth oscillations and speed up training [43].Adam, Adagrad, and RMSProp: These algorithms adapt the learning rate during training for each weight individually, which can lead to quicker convergence [40,58].Optimization algorithms in the learning process of CNNs: This approach enables the adjustment of weights and biases in pretrained CNN models to enhance accuracy in the detection and diagnosis of skin cancer [81].To summarize, the synthesis of CNNs with robust optimization approaches is vital for the success of contemporary deep learning applications [23,29].What types of optimizers are used to improve the accuracy of the results?Throughout the extensive analysis of the literature, a range of algorithms designed for skin cancer detection was evident, featuring a variety of network architectures and tuning parameters and the integration of optimizers within their training routines. These elements are crucial in a CNN framework, as they serve to refine and augment key aspects such as the following:Stability and convergence: Different optimizers possess properties that can impact the stability and rate of convergence during training. Selecting the appropriate optimizer can aid in circumventing issues such as training stagnation or divergence.Efficient learning: Optimizers enable the efficient adjustment of weights and biases in a CNN throughout the training process. This is vital for developing a model that can effectively learn from a dataset of skin images, which is often large and intricate.Overcoming local minima: Optimizers assist in navigating past local minima in the loss function, which is particularly crucial in complex problems like skin cancer detection, where the objective function may have multiple local minima [58].Considering the points discussed, it can be suggested that optimizers are critical to the effectiveness of CNN algorithms for detecting skin cancer [74], as they facilitate efficient training, ensure proper convergence, and enable the model to overcome optimization challenges within the neural network’s weight adjustments. The choice of an optimizer, along with its hyperparameter tuning, is an essential aspect of developing a detection algorithm. In general, from the literature analyzed, Adaptive Moment Estimation (ADAM), stochastic gradient descent (SGD), and root mean square propagation (RMSProp) emerged as the most frequently employed optimizers, Table 8 summarizes the advantages and disadvantages for optimizer mentioned.In the reviewed papers, it was feasible to discern the significance of the outcomes achieved for various algorithm configurations in conjunction with optimizer integration, with ADAM being the most commonly employed one. Presented in Figure 11 is an outline of three selected papers, which illustrate three different stages of contributions within the research domain in recent years. For instance, the first work reported a different design based on CNNs which achieved 68% accuracy, in comparison with the 88% accuracy yielded by hybrid pretrained approaches, which illustrates a relevant improvement.In the first study, entitled “Deep learning outperformed 11 pathologists in the classification of histopathological melanoma images” [31], a deep learning method solely based on CNNs that surpassed 11 pathologists in classifying histopathological melanoma images was highlighted, achieving 68% accuracy. The second paper, “An approach for multiclass skin lesion classification based on ensemble learning” [65], employed an ensemble learning approach to classify skin lesions into multiple categories, utilizing specific algorithms like ResNeXt, SeResNeXt, DenseNet, Xception, and ResNet to achieve an average accuracy of 88% on a dataset of 18,730 dermoscopy images. This result significantly exceeds that of the first study, which solely used CNNs. Finally, in the third article, “Deep learning techniques for skin lesion analysis and melanoma cancer detection: a survey of state-of-the-art” [40], a variety of deep learning techniques for skin lesion analysis and melanoma cancer detection are summarized, highlighting a CNN model optimized with a deep residual network and CDCNN and achieving 99.2% accuracy in classifying 11,720 images from the ISIC 2018 database, providing an overview of advancements in this field up to the year 2020.Overall, the use of optimizers seems to be a consistent strategy for achieving improvements in network performance, with ADAM standing out in several studies, suggesting its popularity and efficacy in optimizing various CNN architectures. However, in some cases, SGD or RMSProp may perform equally well or even better, especially when the hyperparameters are properly adjusted. The final selection often involves experimentation and fine-tuning, meaning the choice of optimizer algorithms depends greatly on the specific problem and the architecture of the neural network.What are the datasets used in skin cancer detection studies?The most commonly used datasets in the reviewed studies were the Human Against Machine (HAM10000) dataset, primarily created by dermatologists at Harvard and other institutions, and the International Skin Imaging Collaboration (ISIC) dataset. Both contain a large number of images (10,015 and 11,720, respectively), compiling a series of dermatoscopic images of skin lesions, including skin cancer characterized by high-quality photographs. In the case of the HAM10000 dataset, seven diagnosed categories are included, with nevus, melanoma, and carcinoma being among the most frequent ones. These images have been extensively used for training machine learning models, specifically for classification tasks through CNNs [46,48]. The ISIC dataset represents a global contribution initiative, featuring a vast collection of images that include various categories but with a focus on melanoma-type cancer [2,82,83,84,85,86].Also, both datasets have been utilized for research and development of automatic diagnostic tools, serving as a standard reference in international challenges and competitions. They include metadata related to clinical diagnosis, lesion type, and body location, among other factors. Their open access and the diversity of data they offer make them highly valuable to the scientific community, in dermatological research, and in the development of artificial intelligence tools for the diagnosis of skin diseases as a significant complement to expert diagnosis [87,88,89,90,91]. Additionally, the PH2 dataset consists of a recompilation of 200 images, focusing on a local objective rather than being broadly applicable to other case studies [69,74]. Regarding well-known issues within the source of the dataset employed, although in the minority, some researchers use non-public databases and internet images, complicating the replication of results due to data unavailability and potential bias in the selection of internet images [92,93,94,95]. On the other hand, most of the datasets currently available focus on skin lesions for lighter skin tones, with many of the images in the ISIC dataset originating mainly from the United States, Europe, and Australia. Furthermore, in order to achieve higher degrees of accuracy and effectiveness when working on the classification, the details transform into key elements to consider, such as training the model while considering the intensity within the color of the skin [43,47]. The size of the lesion also plays a crucial role, as lesions that are smaller than 6 mm tend to be more challenging to identify. As previously addressed in the first question, the treatment or preprocessing of images [33] prior to their training with convolutional neural networks (CNNs) is a critical step for enhancing the efficacy and efficiency of a model. The following key points are suggested for consideration in order to enhance image processing:Quantity–Data augmentation: This involves applying random transformations (such as rotation, translation, scaling, flipping, and brightness adjustments) to the images in the training set. This helps to better generalize the model, making it more robust to variations in new images [96,97].–Generative adversarial networks (GANs): These are a synthetic data generation method that aims to produce samples of images that appear real, referring to a minimax game between two players: a generator and a discriminator. The generator transforms a distribution of random noise into realistic images, while the discriminator learns to differentiate between these generated images and real training data [32].Quality–Color standardization: Ensuring that all images have the same color space (for example, RGB or grayscale) is crucial for maintaining data consistency [98].–Contrast adjustment: Enhancing the contrast of an image can help to highlight important features.Size–Resizing: Images should be of a uniform size before being fed into a CNN. Therefore, it is important to verify the size of the entire dataset and choose one that allows for the preservation of important information without being excessively large, helping to reduce computational requirements [38,57].–Cropping: This is used to cut out parts of the image that do not contain relevant information or to focus on a specific region of the image [38,99].Processing–Normalization: This involves scaling pixel values to have a common range, such as from 0 to 1 or from −1 to 1. This assists the network to train more efficiently. Normalization is typically carried out by dividing the pixel values by 255 (the maximum value for a pixel) [6].–Noise reduction: In some cases, images may contain noise that can be detrimental to the model. Applying filters to reduce or eliminate this noise can be useful [100].Transformation–Whitening: This transforms the image so that it has a mean close to zero and uniform variance. This can enhance convergence during training.–Edge detection and feature extraction: In some cases, it may be useful to preprocess the image to extract specific features, such as edges, using techniques like Sobel or Canny filters. The Sobel filter uses convolutions with two 3 × 3 matrices: one to detect changes in pixel intensity in the horizontal direction (Sobel X) and another for the vertical direction (Sobel Y) [101]. The Canny filter is a more sophisticated approach to edge detection and is considered one of the best due to its accuracy. This process may include various steps and techniques [102]. Both filters have their own strengths, as the Sobel filter is simpler and faster to compute, while the Canny filter is more robust and effective in precise edge detection, especially in the presence of noise [103].The selection of preprocessing techniques is largely contingent upon the nature of the problem and the particular dataset in question [104,105]. In addition to the aforementioned points, the inclusion of clinical data such as race, age, gender, and skin type in classification systems could substantially improve their precision, offering dermatologists valuable supplementary information for decision-making processes. Therefore, it is essential to consider the metadata of the datasets available, conducting an analysis and review of the intersection of information with images. These aspects are crucial for future research in this application domain. Moreover, it has been noted that deep learning tends to be more efficacious than traditional machine learning methods, especially in datasets with a substantial number of images per class. This efficacy can also be replicated in smaller datasets, utilizing data augmentation techniques.What metrics are used to validate the effectiveness of these algorithms in skin cancer detection?

Regarding this question, it is important to mention that in the summary of Table 6, which integrates data from the reviewed studies, various values stand out, mainly corresponding to the metrics of accuracy, recall, and specificity. These metrics are crucial indicators that provide significant information about an algorithm’s performance, and together, they offer a comprehensive overview of the performance of the algorithms analyzed in the documents [106,107,108,109]. This allows for a robust evaluation of their predictive capacity and reliability. However, there are other metrics that can more comprehensively complement the analysis of algorithm results for skin lesion detection. In this context, a set of metrics focused on this area of study, considering its importance in the sensitivity of the data used, should consider the following:Recall: This is primarily for detecting the majority of malignant lesion cases, as failing to identify a melanoma can have severe consequences [110].Specificity: This reduces false positives, which is crucial to avoid unnecessary biopsies and the anxiety associated with a misdiagnosis [111].F1 score: This is important for achieving a balance between sensitivity and precision, especially when working with imbalanced data, as is common in skin lesions where melanoma cases may be less frequent than benign ones [112].AUC-ROC: This provides a comprehensive measure of model performance across all classification thresholds, aiding in selecting the most appropriate threshold for malignant lesion detection [113].

The combination of these metrics can provide a comprehensive assessment that reflects both the algorithm’s ability to detect malignant lesions and its efficiency in ruling out false positives, both of which are critical aspects in the clinical context of dermatology.

## 5. Conclusions

In this work, a systematic literature review was carried out in order to fully understand relevant contributions, designs, strategies, and datasets illustrated for skin cancer detection. In this regard, it is relevant to highlight the contributions of machine learning within the field, where all design-based approaches have demonstrated their effectiveness in skin cancer detection [4,114,115]. Advanced skin cancer detection methods use various data sources and complex features, which have significantly improved the accuracy and predictive capability of models. Incorporating artificial intelligence techniques has enhanced the evaluation metrics of models and provided valuable support to dermatologists and oncologists. Combining automated tools and expert knowledge can lead to faster and more accurate diagnoses, resulting in more effective treatments. These advancements open new avenues for research in teledermatology, providing quality diagnoses even in areas with a shortage of specialists.

Reflecting on the discussion, several avenues for future research emerge, notably in the optimization of hybrid CNN models, effective handling of unbalanced datasets, extending studies to additional databases, and investigating novel network architectures. A promising area involves enhancing feature fusion in deep convolutional neural networks (DCNNs) through the exploration of more efficient techniques, possibly incorporating physics-based methods such as entropy. Furthermore, addressing challenges related to low contrast and signal-to-noise ratios (SNRs) is crucial for improving CNN efficacy in medical images. Advancements in pre- and postprocessing methods, alongside innovative data augmentation strategies, hold significant potential for boosting CNN performance. The application of advanced optimization algorithms during image preprocessing and model training phases could markedly elevate model effectiveness. Additionally, the adoption of adaptive learning schedules and architecture optimization techniques may refine lesion feature extraction and classification within deep networks. In the realm of multimodal image integration, analyzing complementary clinical data, employing generative adversarial networks (GANs), leveraging transfer learning, and integrating text analysis could facilitate more thorough and accurate assessment in skin lesion detection.

Finally, it is crucial to address and mitigate biases in data and models to ensure that AI solutions are fair and equitable. This requires careful selection and analysis of data, as well as ethical consideration in the development and application of these technologies. Interdisciplinary collaboration among engineers, data scientists, dermatologists, and patients is essential to achieve significant and ethically responsible advancements in skin cancer detection using artificial intelligence.

## Figures and Tables

**Figure 1 diagnostics-14-00454-f001:**
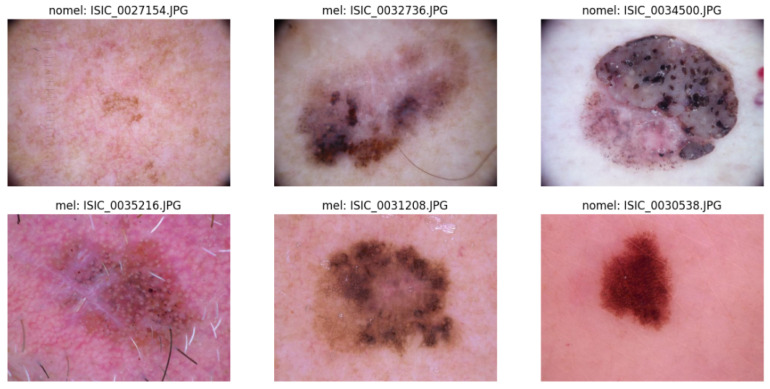
Images from the ISIC 2017 dataset.

**Figure 2 diagnostics-14-00454-f002:**
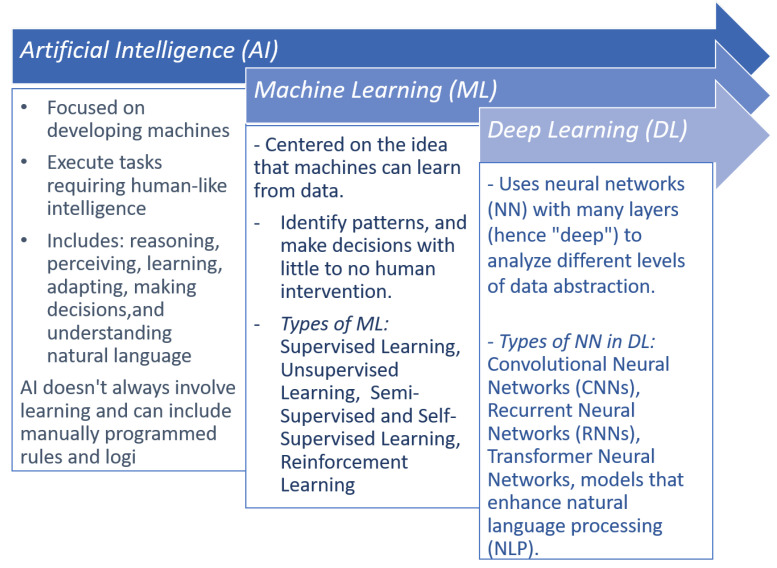
Key aspects of artificial intelligence (AI), machine learning (ML), and deep learning (DL).

**Figure 3 diagnostics-14-00454-f003:**
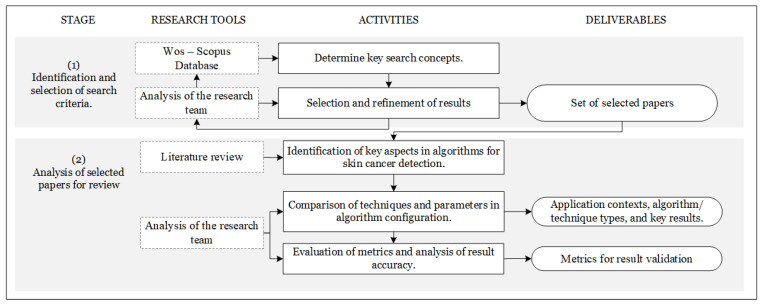
Research methodology.

**Figure 4 diagnostics-14-00454-f004:**
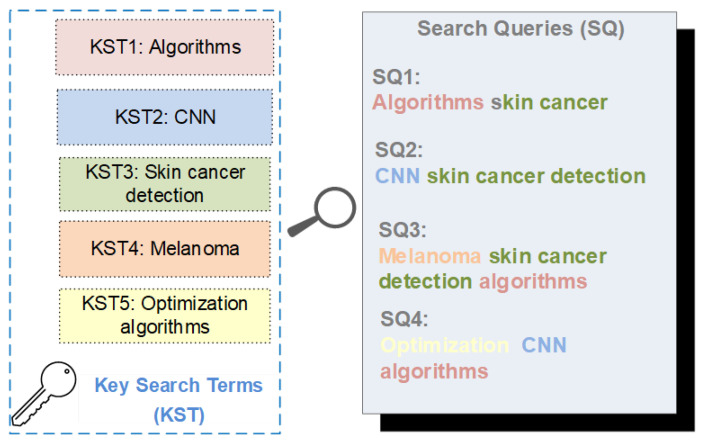
Key terms and search queries.

**Figure 5 diagnostics-14-00454-f005:**
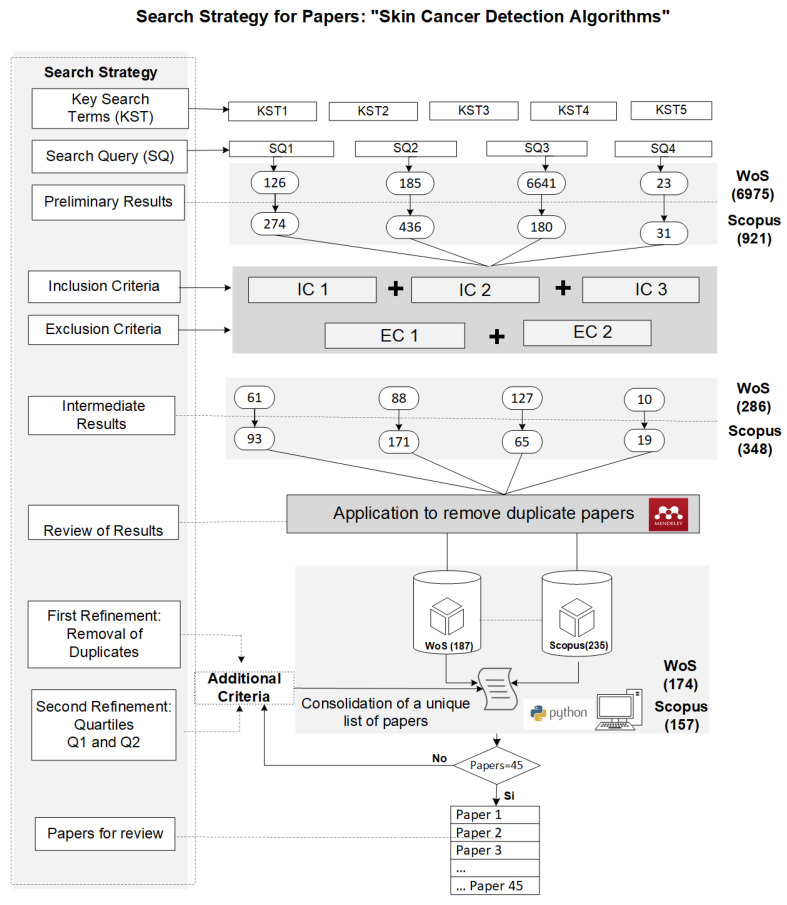
Search strategy based on PRISMA 2020.

**Figure 6 diagnostics-14-00454-f006:**
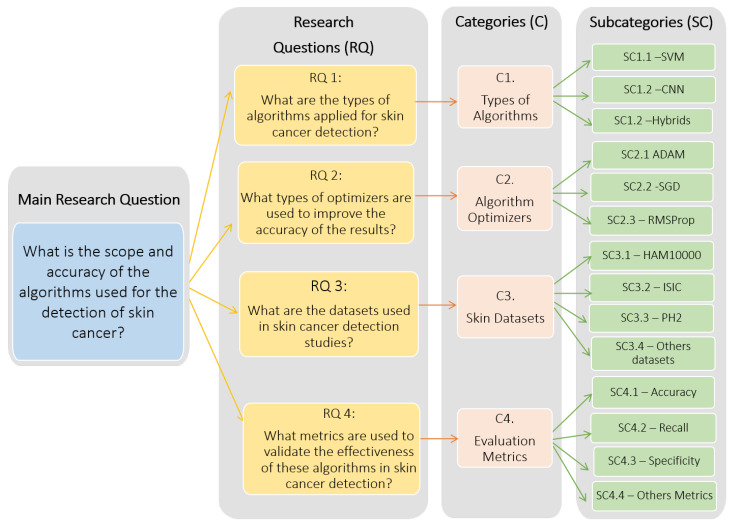
Structure of the systematic review: research questions and categories of analysis.

**Figure 7 diagnostics-14-00454-f007:**
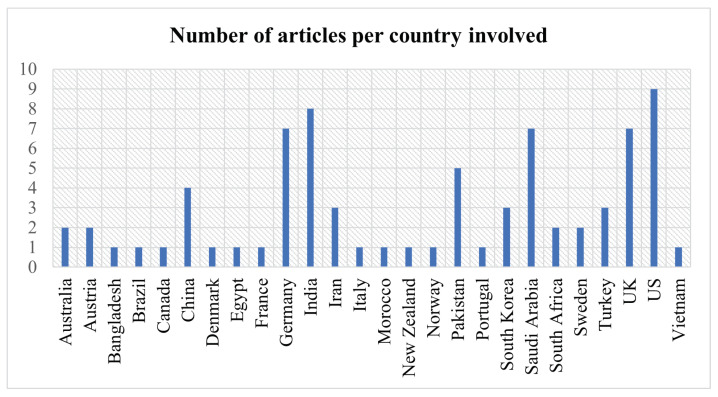
Number of articles per country involved. Note that the total reaches past 45 articles because more than one country can be involved in a study.

**Figure 8 diagnostics-14-00454-f008:**
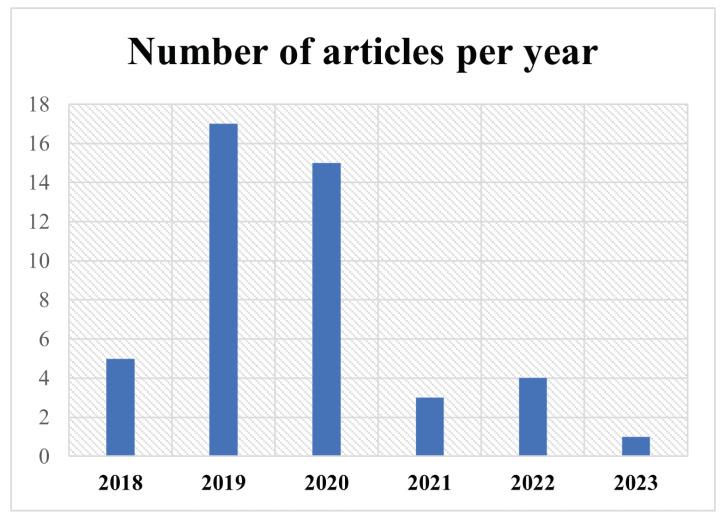
Number of articles per year.

**Figure 9 diagnostics-14-00454-f009:**
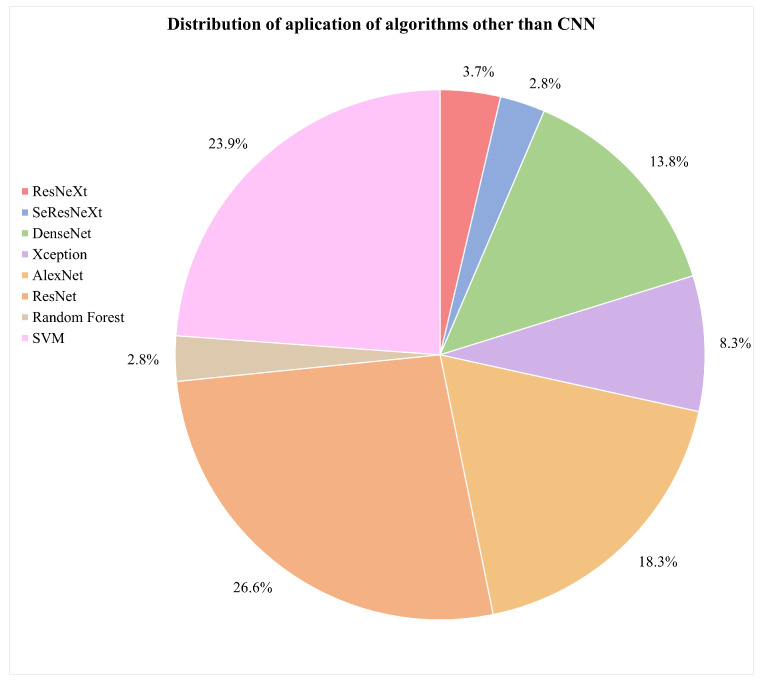
Distribution of application of non-CNN algorithms.

**Figure 10 diagnostics-14-00454-f010:**
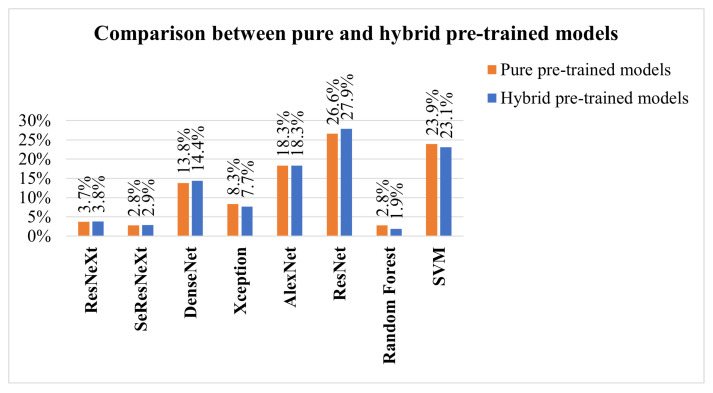
Comparison between pure and hybrid pretrained models.

**Figure 11 diagnostics-14-00454-f011:**
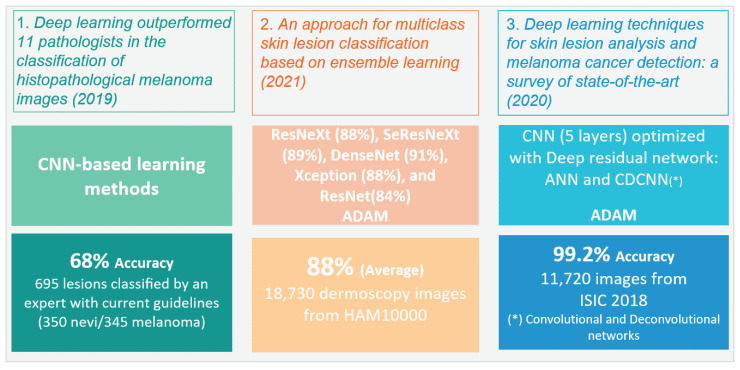
Summary of relevant aspects of some papers.

**Table 1 diagnostics-14-00454-t001:** Detailed descriptions of the ABCDE criteria.

Criteria	Indicates	Description
A	Asymmetry	The majority of illustrated melanomas exhibit an imbalance in their features
B	Border	The borders of melanomas are usually uneven and may have irregular or scalloped edges
C	Color	The presence of multiple colors within a melanoma is a warning sign
D	Diameter	Melanomas tend to be larger, approximately the size of an eraser or around 6 mm in diameter or larger
E	Evolution	This considers any alteration in the shape, size, color, or elevation of a skin spot as a warning sign of melanoma

**Table 2 diagnostics-14-00454-t002:** Inclusion and exclusion criteria.

N°	Criteria: Inclusion (IC) and Exclusion (EC)	Description
IC1	Area of knowledge	Computer science information systems, oncology, computer science artificial intelligence, computer science theory methods, engineering biomedical, computer science software engineering, computer science interdisciplinary applications, medical informatics, multidisciplinary engineering, multidisciplinary sciences
IC2	Language	English
IC3	Document type	Article
EC1	Year	≤2018
EC2	Key words	Systematic review

**Table 3 diagnostics-14-00454-t003:** Summary of the number of papers before and after applying the criteria.

Id SQ	Description	WoS	Scopus
SQ1	Algorithm CNN to skin cancer	126	274
		61	93
SQ2	CNN in skin cancer detection	185	88
		436	171
SQ3	Melanoma cancer detection algorithms	6641	127
		180	65
SQ4	Optimization of CNN algorithms for skin cancer detection	23	10
		31	19
	Preliminary totals	6975	286
		921	348
	Final Results	634	

**Table 4 diagnostics-14-00454-t004:** Summary of the number of papers after applying the refinement criteria.

Refinements	WoS	Scopus
First refinement: duplicate removal	187	235
Second refinement: filtering by quartiles Q1 and Q2	174	157
Final Results	331	

**Table 5 diagnostics-14-00454-t005:** Categories and subcategories of analysis.

Categories	Subcategories	Description
Types of Algorithms	CNN	A convolutional neural network (CNN) is a type of deep neural network specifically designed for the efficient processing of two-dimensional data, such as images and videos. CNNs are inspired by the organization and functioning of the biological visual system, where individual neurons respond to overlapping and superimposed regions of the visual field. This type of network has been particularly successful, especially in the analysis of images in the field of medicine [13].
	SVM	The support vector machine (SVM) is a widely used machine learning model for classification and regression tasks. It is considered a supervised learning algorithm, which means that it is trained using a labeled dataset where the classes or values to be predicted are known [21].
	Hybrids	Other classification algorithms have been identified for skin cancer detection, with deep neural network-based algorithms, particularly recurrent neural networks (RNNs), standing out, as well as a decision tree-based algorithm known as XG-Boost [7,22].
Model Optimizers	ADAM	The ADAM optimization algorithm combines the advantages of stochastic gradient descent (SGD) and the momentum algorithm to dynamically adapt the learning rate. It provides fast and efficient convergence across a wide range of problems [23,24].
	SGD	The stochastic gradient descent is a simple algorithm that updates the network’s weights by using the gradient of the loss function at each iteration. It is the basic optimizer used in many machine learning applications [22].
	RMSprop	RMSprop adapts the learning rate individually for each parameter during training. This means that parameters with large gradients will experience a smaller learning rate, while parameters with small gradients will have a larger learning rate [25].
Skin Datasets	HAM10000	HAM10000 (Human Against Machine) is a dataset consisting of 10,015 dermoscopic images, each of which is a 600 × 450 three-channel RGB image. It provides properly categorized training images [23,26].
	ISIC	ISIC (International Skin Imaging Collaboration) is a publicly available international collaboration dataset of skin images that contains a variety of properly classified skin lesions for research purposes [25].
	PH2	PH2 comprises dermoscopic image databases. PH2 was acquired at the Dermatology Service of Hospital Pedro Hispano in Matosinhos, Portugal, and it includes 200 images, encompassing 40 melanomas and 160 other skin lesions termed nevi (including both atypical and typical nevi) [23].
	Others datasets	Additional datasets, such as the Cutaneous Squamous Cell Carcinoma (cSCC) dataset, involve patients in the study and feature confocal laser scanning microscopy (CLSM) images, with each approximately 10,000 × 10,000 pixels in size [23].
Evaluation Metrics	Accuracy	Accuracy is the ratio of correctly predicted observations to the total observations. For better understanding in the context of the study, it would be the number of images correctly classified—positive and negative—divided by the total number of images [14,27].
	Recall	Recall or sensitivity measures the proportion of actual positives that are correctly identified as such. This is particularly important in medical image processing, where it is crucial to identify as many true cases as possible. In this context, it would be the proportion of images that are correctly identified as belonging to a particular class out of all the images that actually belong to that class [24,28].
	Specificity	Specificity is a metric that refers to the model’s ability to generate responses that are not only accurate but also detailed and relevant to the given context or query. This is particularly important in tasks that require precision and detail-oriented answers. Specificity in this sense is often balanced with other metrics like accuracy, fluency, and relevance [29].
	Others metrics	Metrics such as the receiver operating characteristic (ROC), F1 score, and FNR are crucial in the evaluation of classification algorithms. The ROC curve, along with the area under the curve (AUC), provides a visual and numerical representation of the algorithm’s ability to distinguish between different classes. The F1 score, by merging precision and sensitivity, yields a singular metric that harmonizes these two elements, proving particularly valuable in scenarios with class imbalances. Additionally, precision, also known as the positive predictive value, measures the accuracy of positive predictions. It reflects how many of the items identified as positive are actually positive [23]. The false negative rate (FNR), quantifying the proportion of true positives incorrectly identified as negatives, is critically important in fields where false negatives carry significant consequences, such as in medical diagnostics [24,29].

**Table 6 diagnostics-14-00454-t006:** Findings by analysis category in selected articles.

	Algorithms			Optimizers			Datasets				Metrics			
**ID**	**SVM**	**CNN**	**Hybrids**	**ADAM**	**SGD**	**RMSProp**	**HAM10000**	**ISIC**	**PH2**	**Other Datasets**	**Accuracy**	**Recall**	**Specificity**	**Other Metrics**
Haenssle et al. [30]		x										88.9%	75.7%	x
Brinker et al. [31]		x						x			96%	95%	95.18%	x
Munir et al. [32]		x						x				76%	81.7%	
Al-masni et al. [33]								x						
Saba et al. [34]	x	x						x	**x**	x	98.4%	98.25%	98.5%	x
Goyal et al. [35]		x	x					x	**x**	x	95.67%	92.08%	98.58%	x
Brinker et al. [36]		x			x		x	x				92.8%	68.2%	
Brinker et al. [37]		x						x				82.3%	77.9%	
Mahbod et al. [38]		x	x	x				x			96.3%			x
Hekler et al. [39]		x									68%	76%	60%	
Adegun and Viriri [40]		x	x	x				x		x	99.2%	83.3%	98.6%	x
Zhang et al. [41]		x									97%	93.5%	92%	
Kadampur and Riyaee [42]							x				98.99%			x
Mahbod et al. [43]	x	x	x			x		x						x
Hekler et al. [44]		x												
Ashraf et al. [45]											97.9%			
Maron et al. [46]		x										74.4%	98.8%	
Albahar [47]		x						x			97.49%			x
Kumar et al. [48]	x	x					x		**x**		97.4%			
Nawaz et al. [49]		x						x	**x**	x	95.6%			
Khan et al. [50]	x	x	x							x	96.5%			
Turani et al. [51]	x										98%	97%	98%	x
Dey et al. [52]										x	96.19%	98.41%	91.16%	x
Tan et al. [53]		x	x		x			x	**x**	x				x
Öztürk et al. [54]		x						x	**x**	x	96.92%	96.88%	95.31%	x
Gu et al. [55]		x		x			x				82.9%	58.9%	97.1%	x
Thanh et al. [56]								x			96.6%	96.1%	96.8%	x
Amin et al. [57]	x	x	x					x	**x**	x	99.9%	99.52%	99.62%	x
Bakkouri and Afdel [58]		x		x	x		x				98.09%	93.35%	98.88%	x
Wei et al. [59]	x	x	x	x						x	96.2%	93.9%	97.4%	x
Kaymak et al. [60]		x						x			94.81%			
Khan et al. [61]	x	x							**x**	x	97.74%	97.39%	100%	x
Anand et al. [62]		x	x	x			x				97.96%			
Okur and Turkan [63]		x						x		x	94%			
Abbas and Celebi [64]		x						x				93%	95%	x
Rahman et al. [65]		x	x	x		x	x	x			88%			x
Oskal et al. [66]		x										92.01%		
Sreelatha et al. [67]									**x**		98.64%		99.22%	
Olugbara et al. [68]									**x**	x				
Alizadeh and Mahloojifar [69]	x	x	x		x			x	**x**		97.5%	100%	96.88%	x
Wu et al. [70]		x									87.25%			x
Shetty et al. [71]		x		x			x				95.18%			x
Nasr-Esfahani et al. [72]		x		x	x						95.7%	92.77%	96.3%	x
Mohakud and Dash [73]		x						x			98.33%			
Abunadi and Senan [74]		x						x	x		97.91%			
Presence in the literature review (%)	20%	80%	24%	22%	13%	4%	18%	49%	24%	29%	71%	53%	53%	56%

**Table 7 diagnostics-14-00454-t007:** Datasets and their online access links.

Dataset Name	Link to Dataset (accessed on 23 October 2023)
HAM10000	https://dataverse.harvard.edu/dataset.xhtml?persistentId=doi:10.7910/DVN/DBW86T
ISIC	https://challenge.isic-archive.com/data
PH2	https://www.fc.up.pt/addi/ph2%20database.html

**Table 8 diagnostics-14-00454-t008:** Optimizers.

Optimizer Characteristics	ADAM	SGD	RMSprop
Advantages	– Fast convergence rate in many deep learning applications. – Automatic learning rate adaptation for each parameter. – Combines first and second moment estimations, making it robust against various types of loss functions [72].	– Simple and easy to implement. – Requires less memory compared with more complex algorithms. – Can perform well with properly tuned learning rates.	Adapts the learning rate individually for each parameter, making it effective.
Disadvantages	– May require more memory to store first and second moment estimations. – Can be sensitive to hyperparameters such as the learning rate and exponential decay factors.	– May converge slowly on difficult problems or become trapped in local minima. – Requires fine-tuning of the learning rate.	– May require careful adjustment of the learning rate. – Not always as fast as ADAM in convergence.

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
