# Peer review of "Skin Cancer Detection and Classification Using Neural Network Algorithms: A Systematic Review"

_diagnostics, 2024, doi:10.3390/diagnostics14040454_

Round 1

Reviewer 1 Report

Comments and Suggestions for Authors

I see several problems in this paper

1. The paper is verbose. Its length should be reduced by 15 or 20%. There are repetitions, e.g. the accuracy/recall/specificity are explained at least twice (in the table and in the text). 

2. The English should be improved.

3. The presentation is not always clear. For example it would be better to first introduce the TP,FP,TN and FN and after that the derived metrics (precision etc).

4. A link to the data sets should be added

5. line 149, figure 1 should read figure 4

6. Q1 and Q2 quartiles: not clear, explain better

7. You selected papers based on the number of citations: this will bias the list towards older papers. In fact you have very few papers from 2023. You should use a balancing factors

8. Fig 10, 11 and lines 510 and following: ResNet, AlexNet DenseNet are all CNNs. Why you say these are not CNN ? Figure 11 is particulary obscure.

9. Lines 757 and 758: the conclusion is not supported by the contents: the GANs are never mentioned in the paper

Comments on the Quality of English Language

See previous comments. Should be improved and made concise

Reviewer 2 Report

Comments and Suggestions for Authors

Comments to the Author

In this review, authors have reviewed the available literature involving the use of artificial intelligence in skin cancer detection. It is a good compilation of relevant research post 2018. I recommend addressing the following comments.

1.        The explanation about the layers of skin, although informative, is abrupt and has no link to the manuscript.

2.        Table 6 is not needed and unnecessarily introduces a break in the flow of the manuscript. The table should be removed, and the papers just referenced, or the table should be in supporting information.

3.        Figure 9: This manuscript was submitted in December 2023, but the authors need to mention the date on which the final search of the papers included in the review was performed to better explain the low number of papers in 2023.

4.        Examples or instances of use of AI for disease diagnosis in fields other than skin cancer would help showcase successful application of AI and evolution of different algorithms that have proven successful.

Comments on the Quality of English Language

1.        The manuscript has some awkward sentences that should be addressed. One such example, “Secondly, and considering the aforementioned…..”

Round 2

Reviewer 1 Report

Comments and Suggestions for Authors

Some of my suggestions have been dealt with but the paper is still verbose in my opinion and should be reduced. For example lines 54-109 in the introduction are out sf scope and could be replaced with a single phrase. Table 6 is a repetition of the references: the only additional information conveyed is the country where the paper came from, which could be compressed in a much shorter table or simply eliminated. Overall, the language is verbose more ad-like than scientific-like, so to say. Moreover, the results are not revolutionary or unexpected and could be conveyed in a few pages.

This said, I realise that, to some extent, this is a matter of taste, so I suggest that the editor takes a decision based on his own reading too.

Comments on the Quality of English Language

As said, verbose

Round 3

Reviewer 1 Report

Comments and Suggestions for Authors

Thanks for implementing my requests. I still think that the paper is verbose but we can live with it

Comments on the Quality of English Language

I still think that the paper is verbose but we can live with it